# Association of initiating CYP2D6-metabolized opioids with risks of adverse outcomes in older adults receiving antidepressants: A retrospective cohort study

Yu-Jung Jenny Wei[1*], Almut G. Winterstein[2,3,4], Siegfried Schmidt[5], Roger B. Fillingim[6], Michael J. Daniels[7], Steven T. DeKosky[8], Stephan Schmidt[9]

1 Division of Pharmaceutics and Pharmacology, College of Pharmacy, The Ohio State University, Columbus, Ohio, United States of America, 2 Department of Pharmaceutical Outcomes and Policy, College of Pharmacy, University of Florida, Gainesville, Florida, United States of America, 3 Center for Drug Evaluation and Safety, University of Florida, Gainesville, Florida, United States of America, 4 Department of Epidemiology, Colleges of Medicine and Public Health & Health Professions, University of Florida, Gainesville, Florida, United States of America, 5 Department of Community Health and Family Medicine, College of Medicine, University of Florida, Gainesville, Florida, United States of America, 6 Pain Research and Intervention Center of Excellence, University of Florida, Gainesville, Florida, United States of America, 7 Department of Statistics, College of Liberal Arts and Sciences, University of Florida, Gainesville, Florida, United States of America, 8 Department of Neurology, McKnight Brain Institute, University of Florida, Gainesville, Florida, United States of America, 9 Department of Pharmaceutics, College of Pharmacy, University of Florida, Gainesville, Florida, United States of America

* wei.1342@osu.edu

## Abstract

### Background

The safety of pharmacokinetic opioid-antidepressant interactions may be affected by the sequence in which the drug is initiated. Previous literature showed that initiation of cytochrome P450 (CYP) 2D6-inhibiting versus CYP2D6-neutral antidepressants concomitantly with existing CYP2D6-metabolized opioids (i.e., antidepressant-triggered interaction) was associated with heightened risks of adverse outcomes (e.g., worsening pain). However, little is known about whether and to what extent the risks exist when CYP2D6-metabolized opioids are initiated on existing antide-pressants (i.e., opioid-triggered interaction), a more common pattern of concomitant use of these two drugs. The study aims to examine the association of initiation of CYP2D6-metabolized opioids with risks of adverse outcomes among older nursing home residents who already received antidepressants.

### Methods and findings

We conducted a retrospective cohort study using a 100% Medicare nursing home sample linked to Medicare claims and Minimum Data Set (MDS) assessments from January 1, 2010, to December 31, 2021. Participants included long-term care

**Data availability statement:** Data cannot be shared publicly because the data are licensed to the authors through a data user agreement with the U.S. Centers for Medicare and Medicaid Services. Individual researchers can request access to the data by submitting a request through Research Data Assistant Center at https://resdac.org/. Researchers should be aware that if their request is approved, they will need to purchase a license to access the data. More information is available at https://resdac.org/what-you-need-know-before-you-submit-request.

**Funding:** YJW received grant R01AG073442 from the National Institute on Aging (NIA) (https://www.nia.nih.gov/). NIA had no role in the design and conduct of the study; collection, management, analysis, and interpretation of the data; preparation, review, or approval of the manuscript; and decision to submit the manuscript for publication.

**Competing interests:** The authors have declared that no competing interests exist.

**Abbreviations :** ADL, activities of daily living; aIRR, adjusted incidence rate ratio; aRR, adjusted risk ratio; CYP, cytochrome P450; ED, emergency department; FDA, Food and Drug Administration; IPCW, inverse probability of censoring weighting; IPTW, inverse probability of treatment weighting; IRR, incidence rate ratio; MDS, Minimum Data Set; OD, opioid overdose; ORAEs, opioid-related adverse events; OUD, opioid use disorder; PHQ-9, Patient Health Questionnaire-9; RD, risk difference; SMD, standardized mean difference.

residents 65 years of age or older who initiated CYP2D6-metabolized opioids while already receiving antidepressants for at least 30 days. The key exposure was the use of CYP2D6-inhibiting (versus CYP2D6-neutral) antidepressants concomitantly with CYP2D6-metabolized opioids, with day 1 of antidepressant-opioid concomitant use designated as cohort entry. Patients were followed from cohort entry until the end of 1 year, nursing home discharge, death, or study end (12/31/2021). Seven adverse outcomes included worsening pain, physical function, and depression, and counts of pain-related hospitalizations and emergency department (ED) visits, opioid use disorder (OUD), and opioid overdose (OD). We identified 127,200 older nursing home long-term residents who initiated CYP2D6-metabolized opioids while already receiving antidepressants (mean [SD] age, 84.4 [8.7] years). After covariate adjustment via inverse probability of treatment weighting, use of CYP2D6-inhibiting (versus CYP2D6-neutral) antidepressants concomitantly with CYP2D6-metabolized opioids was associated with a higher risk of worsening pain (relative risk:1.04 [95% CI, 1.02, 1.06]; $P < 0.001$; risk difference (RD): 1.1% [95% CI, 0.6%, 1.6%]) and a higher incidence rate of pain-related hospitalizations (incidence rate ratio [IRR]:1.13 [95% CI, 1.04, 1.22]; $P = 0.003$; RD: 1.21 [95% CI, 0.39, 1.89] per 1,000 patient-years) and pain-related ED visits (IRR = 1.17 [95% CI, 1.07, 1.29]; $P = 0.003$; RD: 0.85 [95% CI, 0.29, 1.41] per 1,000 patient-years), with no difference in physical function, depression, OUD, and OD. Main study limitations included unmeasured confounding and limited generalizability.

## Conclusion

This cohort study of older nursing home residents showed that initiation of CYP2D6-metabolized opioids on existing CYP2D6-inhibiting (versus CYP2D6-neutral) antidepressants was associated with increased risk of worsening pain, pain-related hospitalizations, and pain-related ED visits, although the relative and absolute risks are small to moderate. Clinicians should be aware of potential worsening pain and hospital and ED visits due to pain among patients who used CYP2D6-metabolizing opioids concomitantly with antidepressants, particularly those with CYP2D6-inhibiting antidepressants.

## Author summary

### Why was this study done?

- The safety of pharmacokinetic opioid-antidepressant interactions can be affected by the sequence in which drug is initiated, with the interaction being triggered by initiation of antidepressants on existing CYP2D6-metabolized opioids (i.e., antidepressant-triggered interaction) or by initiation of CYP2D6-metabolized opioids on existing antidepressants (i.e., opioid-triggered interaction).

- While the literature has examined outcomes associated with the opioid-antidepressant interaction triggered by initiation of antidepressants, no studies have yet investigated the interaction triggered by initiation of CYP2D6-metabolized opioids.

## What did the researchers do and find?

- The analysis assessed three clinical outcomes and four opioid-related adverse outcomes during the follow-up (up to 1 year) after the initiation of CYP2D6-metabolized opioids on existing antidepressants among 127,200 older Medicare nursing home long-term residents.

- We found that concomitant use of CYP2D6-inhibiting (versus CYP2D6-neutral) antidepressants with CYP2D6-metabolized opioids was associated with a higher risk of worsening pain, pain-related hospitalization, pain-related ED visits, although the relative risk and absolute risk differences of these outcomes were small to moderate.

- We found no difference in physical function, depressive symptoms, and incident diagnosis of OUD and OD between concomitant users of CYP2D6-inhibiting (versus CYP2D6-neutral) antidepressants with CYP2D6-metabolized opioids.

## What do these findings mean?

- Clinicians should be aware of worsening pain and pain-related hospital and ED visits among patients who used CYP2D6-metabolizing opioids concomitantly with antidepressants, especially those with CYP2D6-inhibiting antidepressants.

- The study findings are only generalizable to older nursing home long-term residents and are limited by not fully accounting for confounders, particularly those that cannot be measured in the Medicare and MDS data.

## Introduction

Opioids are a major component of comprehensive pain treatment for older adults with chronic pain, including those living in nursing homes. Over 90% of older nursing home residents with opioid therapy received opioids (i.e., hydrocodone, codeine, tramadol, and oxycodone) metabolized in the liver by the cytochrome P450 (CYP)2D6 enzyme for managing pain [1]. Because CYP2D6 is a major metabolic enzyme of these opioids to active or potent analgesic metabolites [2,3], concomitant use of CYP2D6-metabolized opioids with medications that inhibit the CYP2D6 enzyme, such as certain antidepressants, may cause inadequate analgesia. The reduction in analgesic effects associated with opioid-antidepressant interaction results in worsening of pain [4–8], and persistent pain can lead to increased risk for impaired physical functioning and mental health problems, including depression [9]. The reduction in analgesic effects can also result in opioid-related adverse events (ORAEs), including pain-related hospitalizations and emergency department [ED] visits) [4,8,10].

The safety of pharmacokinetic opioid-antidepressant interactions can be affected by the sequence in which drug is initiated. The interactions can be triggered by initiation of CYP2D6-inhibiting antidepressants among patients who already received CYP2D6-metabolized opioids (i.e., antidepressant-triggered interaction). Because this group already used opioids, antidepressant-triggered interactions may precipitate opioid withdrawal symptoms from decreased analgesia of opioids, heightening risks of not only worsening pain but also pain-related medical encounters, as demonstrated in prior studies [8,10]. On the other hand, the interactions can be triggered by initiation of CYP2D6-metabolized opioids among patients who already received antidepressants (i.e., opioid-triggered interaction). When patients initiate

CYP2D6-metabolized opioids while still on antidepressants, they may encounter adverse events (e.g., worsening pain) likely due to decreased analgesia from the antidepressant-opioid interaction but have no immediate concern for opioid withdrawal symptoms.

Today, little is known whether and to what extent the risks of adverse outcomes are associated with pharmacokinetic antidepressant-opioid interactions triggered by initiation of CYP2D6-metabolized opioids among older nursing home residents. Our prior study has examined adverse outcomes associated with the interactions triggered by initiation of antidepressants [8]; however, these patients constitute only 20% of older nursing home patients who concomitantly used both CYP2D6-metabolized opioids and antidepressants according to our preliminary data using the 100% nursing home sample [8]. The predominant (80%) concomitant users of both drugs in nursing homes are patients who already received antidepressants followed by initiation of CYP2D6-metabolized opioids, which has not yet been studied.

Using a national nursing home sample, we aimed to study the safety of pharmacokinetic antidepressant-opioid interactions triggered by initiation of CYP2D6-metabolized opioids among older patients who were already receiving antidepressants. Following our prior study [8], we compared three key clinical outcomes (i.e., pain intensity, physical function, and depression) and four ORAE outcomes (i.e., opioid-related hospitalizations and ED visits, opioid use disorder [OUD], and opioid overdose [OD]) among patients with CYP2D6-inhibiting versus CYP2D6-neutral antidepressants, concomitantly used with newly-prescribed CYP2D6-metabolized opioids. We hypothesize that concomitant use of CYP2D6-inhibiting (versus CYP2D6-neutral) antidepressants with CYP2D6-metabolized opioids was associated with risks of clinical and ORAE outcomes.

## Methods

### Study source and design

This retrospective cohort study analyzed data from a 100% nursing home resident sample linked to Medicare claims and Minimum Data Set (MDS) assessments from January 1, 2010, to December 31, 2021. Medicare claims data contain fee-for-service beneficiaries' medical billing records for Parts A (inpatient), B (office-based visits), and D (prescription drugs). We used Medicare Part D data to measure concomitant antidepressant-opioid use (key exposure) and used Parts A and B to ascertain key ORAE outcomes. The MDS 3.0 assessment is required at admission, regular intervals during a Medicare-covered short-term stay, and quarterly thereafter [11]. We used MDS 3.0 assessments to measure key clinical outcomes (pain intensity, physical function, and depressive symptoms) and important covariates, such as cognitive function, body mass index, and use of as-needed (PRN) pain medications among residents with a long-term stay (>100 days). [12] The Ohio State University's Institutional Review Board approved this study. This study followed the STROBE reporting guideline (S1 STROBE Checklist). A prospective statistical analysis plan was developed (S1 Text).

### Study sample

The study sample included older (≥65 years) long-term residents who have CYP2D6-metabolized opioids initiated while already receiving antidepressants for at least 30 days between January 1, 2011 and December 31, 2020. Cohort entry was defined as day 1 of antidepressant and opioid concomitant use. Initiation of CYP2D6-metabolized opioids was defined as having no such prescription filled within 180 days before cohort entry (i.e., baseline). To minimize confounding by disease conditions, we restricted the study sample to those with a diagnosis of chronic pain for opioid therapy initiation within 1 month and conditions that are approved by the US Food and Drug Administration (FDA) or off-label use for antidepressants within 6 months before the cohort entry. Residents were followed up from cohort entry until the end of 1-year follow-up, Medicare disenrollment, death, nursing home discharge, or study end (December 31, 2021).

To ensure comparability with our prior study that examined opioid-antidepressant interactions triggered by initiation of antidepressants [8], we used the same inclusion and exclusion criteria for the present study that examined the interactions

triggered by initiation of CYP2D6-metabolized opioids. Specifically, to ensure complete capture of baseline medical and prescription claims and MDS data for measuring key variables of interest, we required residents to have (1) continuous enrollment in Medicare Parts A, B, and D, without a Medicare Advantage plan; (2) no hospital or skilled nursing facility stay, during which Medicare-covered services are unavailable due to CMS's prospective payment system policy [13]; (3) ≥1 MDS assessment; (4) continuous nursing home stay; (5) no diagnosis of cancer or palliative or hospice care; and (6) no coma or severe cognitive impairment (defined as having a Cognitive Performance Scale score of 5 or 6) [14], assessed using the MDS assessments, during 6 months before cohort entry. The last criterion was used because clinical pain and depression symptoms among residents with coma or severe cognitive impairment tend to be under-detected [15], and their risk for ORAEs is rare.

The final sample was used for analyses of ORAE outcomes, and a subset of the final sample who had ≥1 MDS assessment in follow-up was used to measure clinical outcome changes. Residents were allowed to re-enter the cohort after the end of follow-up if they met the eligibility criteria. Fig 1 shows the sample selection details. The medications of interest and diagnostic and procedure codes for conditions and services considered in the sample selection are given in S1 and S2 Tables.

### Key exposure

The key exposure of interest was the use of CYP2D6-inhibiting (versus with CYP-neutral) antidepressants concomitantly used with CYP2D6-metabolized opioids that overlapped for at least 1 day during the follow-up. Antidepressants that moderately or strongly inhibit CYP2D6 enzyme (i.e., fluoxetine, paroxetine, duloxetine, doxepin, and bupropion) were classified as CYP2D6-inhibiting antidepressants; otherwise, as CYP-neutral antidepressants (S1 Table). The classification of antidepressants was made based on multiple sources of drug interactions, including the US FDA [16], Micromedex, Lexicomp, and Drug Interaction Database (DIDB) [17]. For each drug, we determined days of drug use according to days' supply and dispending date of prescriptions. We then constructed episodes of concomitant opioid-antidepressant use by identifying the continuous number of overlapping days on which a CYP2D6-metabolized opioid was used along with an antidepressant drug. When constructing episodes of concomitant opioid-antidepressant use, we allowed a gap of fewer than 15 days to account for delays in fills of prescription drugs of interest.

### Adverse outcomes

Following our prior study [8], we studied three clinical measures—worsening pain intensity, physical function, and depression status, all of which were obtained from MDS 3.0 and 4 ORAE measures—pain-related hospitalization and ED visits, OUD, and OD, all of which were assessed using Medicare claims data (S2 Table). Pain-related hospitalization and ED visits were defined as encounters with a primary or secondary diagnosis of a chronic pain condition. OUD and OD were captured from inpatient or outpatient encounters with a diagnosis code for opioid misuse, dependence, or poisoning [18]. For each outcome, we included only patients with no prior events during the 6-month baseline and calculated incident rates of residents who experienced new events over person-days of follow-up.

In MDS 3.0, pain intensity was measured by residents' recall of the worst pain in the last 5 days based on a 10-point numerical rating scale (higher scores indicating greater pain) or a 4-level categorical verbal descriptor scale (no, mild, moderate, or severe pain) [11]. To facilitate analysis, scores of the two pain scales were combined and classified into four categories: no (0), mild (1–4), moderate (5–7), and severe (8–10) pain [19,20]. Physical function was measured by the 9-item (i.e., bed mobility, transfer, walking in room, walking in corridor, locomotion on unit, dressing, eating, toilet use, and personal hygiene) self-care activities of daily living (ADL), with each item scored from 0 (total independence) to 4 (total dependence) [21]. The total ADL score ranged from 0 to 36, with higher scores indicating worse physical function; a cut-off ≥19 was used to define moderate-to-severe physical function. Depression status was measured using the Patient

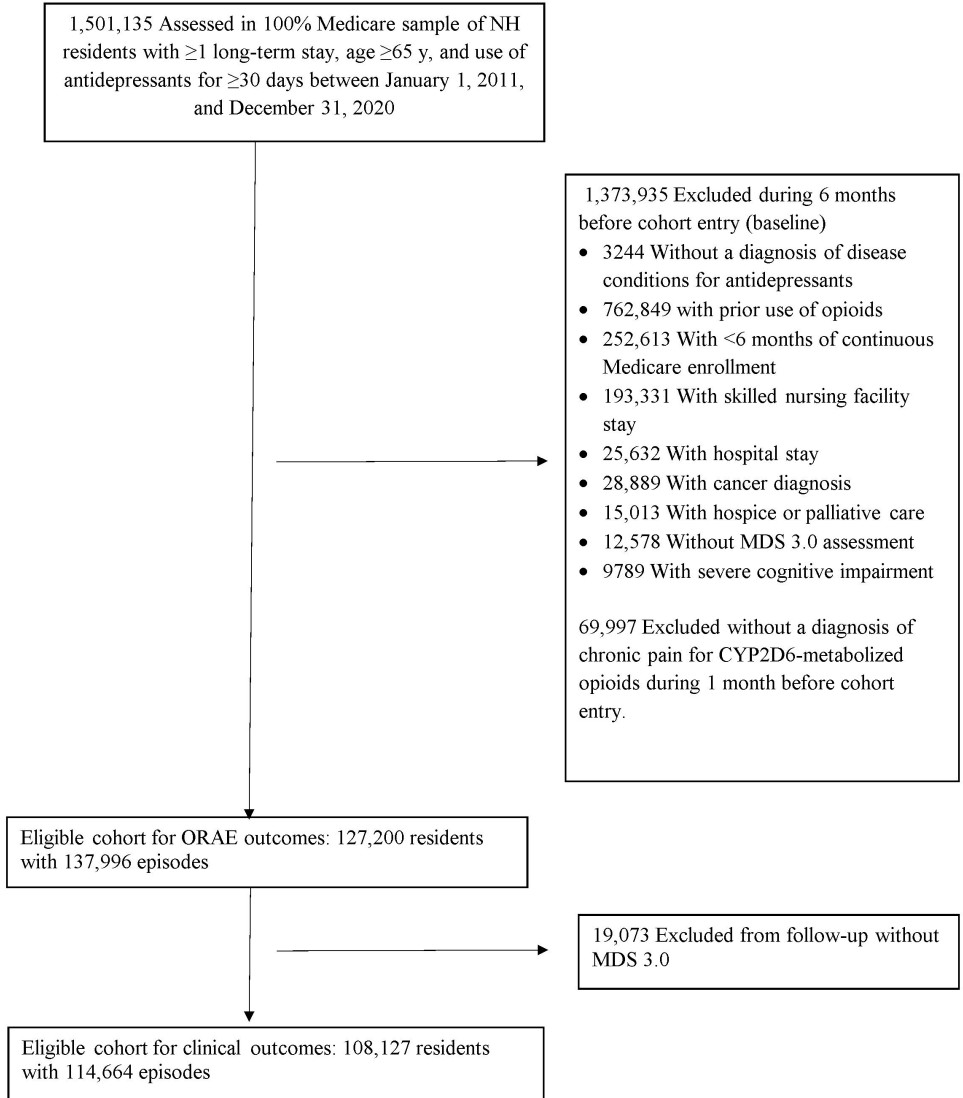

**Fig 1. Cohort inclusion flowchart for the study sample.** NH represents nursing home; CYP, cytochrome P450; MDS, Minimum Data Set; ORAE, opioid-related adverse event.

Health Questionnaire-9 (PHQ-9), with each of the 9 items scored from 0 (not at all) to 3 (nearly every day). The total score ranged from 0 to 27, with higher scores indicating worse depression; a cut-off ≥10 was used to define moderate-to-severe depression [22].

To analyze associations with clinical outcomes mostly obtained from quarterly MDS 3.0 data, we divided patients' follow-up into quarterly intervals. In each patient quarter, we calculated individual changes in outcomes by subtracting each patient's pain, ADL, and PHQ-9 scores from the corresponding baseline outcome extracted from the baseline MDS assessment. The continuous score changes of these symptoms included scores >0 (worsening), <0 (improving), and 0 (no change). Patients with no change in symptoms include two clinically different patient subgroups—(1) patients who remained no-to-mild from baseline to follow-up, suggesting their symptoms were controlled; and (2) patients who remained moderate-to-severe from baseline to follow-up, suggesting their symptoms were uncontrolled or got worse.

Thus, similar to previous studies [8,23], we classified residents with score changes >0 or equal to 0 that remained moderate to severe as having worsening outcomes, and otherwise as improving outcomes (score changes <0 or score changes equal to 0 that remained no-to-mild symptoms).

## Covariates

Covariates measured during 6 months before the cohort entry included demographic characteristics (age, sex, race and ethnicity, region of the United States, and receipt of low-income subsidy), health status (body mass index, and diagnosis of tobacco, alcohol, or drug use disorder), select clinical conditions potentially affecting opioid and antidepressant treatment [24], the total number of comorbidities, pain management, medication use, and nursing home characteristics (S3 Table). Pain management included the use of drug or nondrug pain management, use of PRN pain medications, receipt of procedures and therapies for chronic pain management, use of prescription nonopioids, and use of adjuvant analgesics. Medication use included use of other CNS medications, use of other CYP2D6 inhibitors [2,16,25], use of CYP3A4 inhibitors [2,16,25], use of CYP2D6 inducers [2,16,25], and polypharmacy (use of ≥5 dispensed generic drugs excluding opioids and antidepressants). We accounted for use of other CYP2D6 inhibitors because they may compete with CYP2D6-inhibiting antidepressants for the CYP2D6 enzyme [26], use of CYP3A4 inhibitors because the studied opioids are also metabolized at least to some extent via the CYP3A4 enzyme, which could also affect pain-related outcomes [2,25], and use of CYP2D6 inducers because they increase the metabolic activity of the CYP2D6 enzyme, which can also result in inadequate analgesia and affect pain-related outcomes [2,25]. To account for the potential cluster effect at the facility level, we adjusted for five nursing home characteristics derived from LTCFocus [27], including total number of beds, profit status, chain membership, having special care unit, and geography (metropolitan, micropolitan, or rural, determined based on rural-urban continuum codes). We also accounted for baseline clinical outcomes, cognitive function, duration of antidepressant use, and year of cohort entry.

## Missing data

A small percentage (3.9%) of residents with missing data for the MDS-assessed covariates at baseline were excluded from analyses for clinical and adverse outcomes. For the clinical outcome analysis, residents with missing MDS-assessed clinical outcomes (<1%) during periods of concomitant opioid-antidepressant use in follow-up were also excluded.

## Statistical analysis

We assessed baseline covariates between users of CYP2D6-inhibiting antidepressants (study group) versus CYP-neutral antidepressants (comparison group), concomitantly used with CYP2D6-metabolized opioids. A standardized mean difference (SMD) higher than 0.100 indicates a covariate imbalance [28]. Potential differences in baseline variables between groups were then adjusted via the inverse probability of treatment weighting (IPTW), calculated as the inverse of the propensity score for the study group and the inverse of 1 minus the propensity score for the comparison group. Propensity score was estimated using a logistic regression that modeled the probability of being assigned to the study versus comparison group as the dependent variable, and baseline covariates as the independent variables [29]. Each IPTW value was truncated at the 1st and 99th percentiles to reduce the influence of outliers on estimates. We used ITPW because of its advantages in retaining all study participants and reporting balance in baseline characteristics between the study and comparing groups [30].

For each clinical outcome, we used a robust Poisson regression model [31] to estimate the adjusted risk ratio (aRR) and its 95% confidence interval (CI) of a worsening outcome from baseline to quarterly follow-up assessments between the study versus comparison groups. In each model, we controlled baseline variables via IPTW and quarterly time as a covariate and incorporated a generalized estimating equation to account for within-individual correlation due to repeated

PLOS Medicine

clinical outcome measures. For ORAE count outcomes, we used a Poisson or negative binomial model based on the presence of overdispersion [32] with ITPW to model the adjusted incidence rate ratio (aIRR) and its 95% CI of an adverse outcome among the study versus comparison groups. Days of follow-up were included as an offset variable in ORAE count models. We also calculated the adjusted absolute risk of the study and treatment group and the adjusted risk difference (RD) between groups for each clinical and ORAE outcome using a SAS macro [33]. SAS codes for performing outcome regression analyses were provided in S1 Text.

We performed sensitivity analyses to assess the robustness of the estimates. First, to additionally account for censoring and competing risk of death that differs between the study and treatment group, we included inverse probability of censoring weighting (IPCW), estimated using logistic regression with the censoring due to death as the dependent variable and baseline covariates as independent variables [34,35], along with IPTW in analytical outcome models. Second, to understand whether associations varied by antidepressant classes according to whether they had additional analgesic effects (i.e., serotonin and norepinephrine reuptake inhibitors and tricyclic antidepressants) or not (selective serotonin reuptake inhibitors (SSRIs) and other antidepressants), we stratified analysis for these two groups. Third, to understand whether associations varied by CYP2D6-metabolized opioids, we stratified by specific opioid drugs. Fourth, to minimize potential competing effects arising from the use of other CYP2D6 inhibitors, we restricted the analysis to patients without use of other CYP2D6-inhibiting medications at baseline. Fifth, to address the potential non-independence within individual observations due to cohort re-entry, we included the first eligible observation of each study sample. All analyses were conducted using SAS, version 9.4 (SAS Institute ). Statistical significance was set as $P < 0.05$, and all tests were two-sided.

## Results

We identified 127,200 eligible Medicare nursing home long-term residents who initiated CYP2D6-metabolized opioids while already receiving antidepressants (mean [SD] age, 84.4 [8.7] years; 97,787 [76.9%] female), consisting of 137,996 resident observations from January 2011 to December 2021 (Fig 1 and Table 1). Within this sample, we further identified a subset of 108,127 residents who had ≥1 MDS 3.0 in follow-up (mean [SD] age, 84.0 [8.8] years; 82,650 [76.4%] female), consisting of 114,664 patient episodes for analysis of changes in clinical outcomes (Fig 1 and S4 Table). Among the 127,200 eligible sample, 7.1% of residents were censored due to Medicare disenrollment, 11.8% due to nursing home discharge, and 17.5% due to death. We observed similar proportions of residents censored due to Medicare disenrollment (7.3% versus 7.2%) and nursing home discharge (11.5% versus 11.8%), but different proportions due to death (13.8% versus 17.9%) between the study and comparison group, which was adjusted via the IPCW as a sensitivity analysis.

Of 137,996 resident observations that had concomitant antidepressant-opioid use, 23,856 residents (17.3%) initiated CYP2D6-metabolized opioids while already receiving CYP2D6-inhibiting antidepressants (study group), whereas 114,140 residents (82.7%) initiated CYP2D6-metabolized opioids while already receiving CYP2D6-neutral antidepressants (comparison group). The baseline characteristics are given in Table 1. After IPTW, the distributions of all measured baseline characteristics were well balanced between groups, with SMDs for characteristics <0.100.

### Clinical and ORAE outcomes

For clinical outcomes, the study versus comparison group had a higher crude percentage of residents who experienced any worsening pain (25.5% versus 23.0%), but lower or similar crude percentages of residents with worsening physical function (63.8% versus 65.4%) and depressive symptoms (29.6% versus 29.9%) (Table 2) in follow-up. After covariate adjustment, the study versus comparison group had a higher risk of worsening pain (aRR, 1.04 [95% CI, 1.02, 1.06]; $P < 0.001$) with an absolute increase in the risk of worsening pain by 1.1% (95% CI, 0.6%, 1.6%) during the 1-year follow-up. In contrast, the study versus comparison group had no difference in worsening physical function (aRR, 1.00 [95% CI, 0.99, 1.01]; $P = 0.49$; RD, −0.2% [95% CI, −0.9%, 0.5%]).) and depressive symptoms (aRR, 1.01 [95% CI, 0.99, 1.03];

PLOS Medicine

**Table 1. Clinical and demographic characteristics of nursing home residents receiving concomitant antidepressant-opioid use overall, and by CYP2D6-inhibiting vs. CYP2D6-neutral antidepressants.**

| Characteristic[a] | No. (%) of residents with concomitant antidepressant- opioid use | | | SDiff[b] | |
|---|---|---|---|---|---|
| | Overall sample (*n* = 127,200) | CYP2D6-inhibiting ADs with CYP2D6-metabolized opioids (*n* = 23,856)[c,d] | CYP2D6-neutral ADs with CYP2D6-metabolized opioids (*n* = 114,140)[c,d] | Before IPTW | After IPTW |
| **Age, y** | | | | | |
| Mean (SD) | 84.4 (8.7) | 82.1 (8.9) | 84.4 (8.8) | 0.178 | 0.017 |
| 65-74 | 20,209 (15.9) | 5,607 (23.5) | 18,733 (16.4) | 0.091 | 0.016 |
| 75-84 | 38,073 (29.9) | 7,975 (33.4) | 33,354 (29.2) | 0.228 | 0.065 |
| ≥85 | 68,918 (54.2) | 10,264 (43.0) | 62,053 (54.4) | 0.019 | 0.018 |
| **Female** | 97,787 (76.9) | 18,090 (75.8) | 87,466 (76.6) | 0.178 | 0.015 |
| **Race and ethnicity** | | | | | |
| White | 107,058 (84.2) | 20,241 (84.8) | 95,432 (83.6) | 0.034 | 0.007 |
| Black | 11,312 (8.9) | 1911 (8.0) | 10,515 (9.2) | 0.043 | 0.009 |
| Other[e] | 8,830 (6.9) | 1,704 (7.1) | 8,193 (7.2) | 0.001 | 0.000 |
| **Received low-income subsidy** | 108,546 (85.3) | 20,741 (86.9) | 97,362 (85.3) | 0.048 | 0.009 |
| **US Region** | | | | | |
| Northeast | 24,074 (18.9) | 3,935 (16.5) | 21,827 (19.1) | 0.069 | 0.006 |
| Midwest | 35,778 (28.1) | 6,890 (28.9) | 31,764 (27.8) | 0.023 | 0.004 |
| South | 55,725 (43.8) | 10,331 (43.3) | 50,571 (44.3) | 0.020 | 0.003 |
| West | 11,623 (9.1) | 2,700 (11.3) | 9,978 (8.7) | 0.086 | 0.007 |
| **Body mass index** | | | | | |
| Underweight | 7,632 (6.0) | 978 (4.1) | 7,230 (6.3) | 0.101 | 0.001 |
| Normal weight | 41,424 (32.6) | 5,951 (24.9) | 38,418 (33.7) | 0.192 | 0.012 |
| Overweight | 37,915 (29.8) | 7,039 (29.5) | 33,975 (29.8) | 0.006 | 0.005 |
| Obese | 40,229 (31.6) | 9,888 (41.4) | 34,517 (30.2) | 0.235 | 0.007 |
| **Clinical condition** | | | | | |
| Tobacco or alcohol use disorder | 3,624 (2.8) | 809 (3.4) | 3,311 (2.9) | 0.028 | 0.003 |
| Chronic pain | | | | | |
| Musculoskeletal pain | 121,995 (95.9) | 22,468 (94.2) | 109,667 (96.1) | 0.088 | 0.003 |
| Neuropathic pain | 16,444 (12.9) | 4,317 (18.1) | 14,124 (12.4) | 0.160 | 0.005 |
| Idiopathic pain | 13,554 (10.7) | 3,517 (14.7) | 11,507 (10.1) | 0.148 | 0.004 |
| Mental health disorder | 95,503 (75.1) | 18,323 (76.8) | 85,610 (75.0) | 0.042 | 0.003 |
| Sleep disorder | 18,920 (14.9) | 3,570 (15.0) | 17,285 (15.1) | 0.005 | 0.006 |
| Behavioral symptoms of ADRD | 91,826 (72.2) | 15,605 (65.4) | 83,436 (73.1) | 0.167 | 0.022 |
| Diabetes | 55,258 (43.4) | 11,816 (49.5) | 48,812 (42.8) | 0.136 | 0.004 |
| Cardiovascular disease | 99,270 (78.0) | 18,619 (78.0) | 89,055 (78.0) | 0.001 | 0.003 |
| Hypertension | 106,891 (84.0) | 20,188 (84.6) | 95,794 (83.9) | 0.019 | 0.001 |
| Pulmonary condition | 71,935 (56.6) | 13,871 (58.1) | 64,217 (56.3) | 0.038 | 0.003 |
| Kidney disease | 32,384 (25.5) | 6,357 (26.6) | 29,055 (25.5) | 0.027 | 0.004 |
| Liver disease | 2,844 (2.2) | 584 (2.4) | 2,584 (2.3) | 0.012 | 0.003 |
| Gastrointestinal tract disorder | 51,699 (40.6) | 9,686 (40.6) | 46,496 (40.7) | 0.003 | 0.006 |
| Injury | 45,645 (35.9) | 8,145 (34.1) | 41,119 (36.0) | 0.040 | 0.002 |
| Neurodegenerative disorder | 20,013 (15.7) | 4,239 (17.8) | 17,741 (15.5) | 0.060 | 0.004 |
| Seizure | 11,056 (8.7) | 2,150 (9.0) | 10,163 (8.9) | 0.004 | 0.010 |
| Drug use disorder | 670 (0.5) | 173 (0.7) | 582 (0.5) | 0.028 | 0.001 |

*(Continued)*

| Characteristic[a] | No. (%) of residents with concomitant antidepressant- opioid use | | | SDiff[b] | |
| | Overall sample (*n* = 127,200) | CYP2D6-inhibiting ADs with CYP2D6-metabolized opioids (*n* = 23,856)[c,d] | CYP2D6-neutral ADs with CYP2D6-metabolized opioids (*n* = 114,140)[c,d] | Before IPTW | After IPTW |
|---|---|---|---|---|---|
| Total No. of comorbidities, Mean (SD) | 19.7(7.1) | 20.1 (7.3) | 19.7 (7.1) | 0.062 | 0.003 |
| **Pain management** | | | | | |
| Drug or nondrug pain intervention | 76,849 (60.4) | 15,502 (65.0) | 67,888 (59.5) | 0.114 | 0.002 |
| Any procedure or therapy for chronic pain management | 75,366 (59.3) | 14,469 (60.7) | 67,519 (59.2) | 0.031 | 0.009 |
| Use of other pain medication | | | | | |
| Any adjuvant analgesic | 51,690 (40.6) | 14,693 (61.6) | 42,279 (37.0) | 0.507 | 0.014 |
| Any prescription nonopioid | 18,720 (14.7) | 4,098 (17.2) | 16,368 (14.3) | 0.078 | 0.006 |
| Use of PRN pain medication | 35,726 (28.1) | 7,615 (31.9) | 31,349 (27.5) | 0.098 | 0.004 |
| **Medication use** | | | | | |
| Use of other CNS medication | 120,544 (94.8) | 19,839 (83.2) | 110,871 (97.1) | 0.482 | 0.001 |
| Polypharmacy | 122,246 (96.1) | 23,151 (97.0) | 109,547 (96.0) | 0.058 | 0.000 |
| Use of CYP2D6 inhibitors | 1,034 (0.8) | 1,060 (4.4) | 4,186 (3.7) | 0.039 | 0.000 |
| Use of CYP3A4 inhibitors | 34,202 (26.9) | 6,671 (28.0) | 30,251 (26.5) | 0.033 | 0.006 |
| Use of CYP2D6 inducers | 4,933 (3.9) | 1,002 (4.2) | 4,457 (3.9) | 0.015 | 0.004 |
| **Days of AD use**, Mean (SD) | 126.8 (77.5) | 170.7 (24.4) | 169.3 (26.8) | 0.055 | 0.052 |
| **Baseline Cognitive Function** | | | | | |
| Intact | 34,305 (27.0) | 8,266 (34.6) | 29,681 (26.0) | 0.189 | 0.015 |
| Mild | 38,876 (30.6) | 7,767 (32.6) | 34,454 (30.2) | 0.051 | 0.006 |
| Moderate | 54,019 (42.5) | 7,823 (32.8) | 50,005 (43.8) | 0.228 | 0.019 |
| **Baseline physical dependence** | | | | | |
| No (ADL ≤ 9) | 21,135 (16.6) | 4,199 (17.6) | 18,901 (16.6) | 0.028 | 0.011 |
| Mild (10 ≤ ADL ≤ 18) | 27,030 (21.3) | 4,940 (20.7) | 24,355 (21.3) | 0.016 | 0.002 |
| Moderate (19 ≤ ADL ≤ 27) | 55,828 (43.9) | 10,503 (44.0) | 49,881 (43.7) | 0.007 | 0.004 |
| Severe (28 ≤ ADL) | 23,207 (18.2) | 4,214 (17.7) | 21,003 (18.4) | 0.019 | 0.004 |
| **Baseline depression status** | | | | | |
| No (PHQ-9 ≤ 4) | 98,989 (77.8) | 18,403 (77.1) | 89,029 (78.0) | 0.021 | 0.001 |
| Mild (5 ≤ PHQ-9 ≤ 9) | 17,257 (13.6) | 3,326 (13.9) | 15,337 (13.4) | 0.015 | 0.000 |
| Moderate (10 ≤ PHQ-9 ≤ 14) | 5,552 (4.4) | 1,130 (4.7) | 4,892 (4.3) | 0.022 | 0.001 |
| Severe (15 ≤ PHQ-9) | 1,546 (1.2) | 304 (1.3) | 1,390 (1.2) | 0.005 | 0.002 |
| Missing | 3,856 (3.0) | 693 (2.9) | 3,492 (3.1) | 0.009 | 0.001 |
| **Baseline pain status** | | | | | |
| No | 86,608 (68.1) | 15,079 (63.2) | 78,754 (69.0) | 0.123 | 0.005 |
| Mild | 16,772 (13.2) | 3,411 (14.3) | 13,762 (12.1) | 0.066 | 0.001 |
| Moderate | 14,590 (11.5) | 3,254 (13.6) | 12,623 (11.1) | 0.079 | 0.006 |
| Severe | 5,250 (4.1) | 1,221 (5.1) | 4,508 (3.9) | 0.056 | 0.002 |
| Missing | 4,980 (3.9) | 891 (3.7) | 4,493 (3.9) | 0.011 | 0.001 |
| **NH Characteristics** | | | | | |
| Total number of NH beds, Mean (SD) | 126.8 (77.5) | 124.7 (76.7) | 126.9 (77.2) | 0.030 | 0.003 |
| Chain membership | 66,999 (52.7) | 12,553 (52.6) | 60,132 (52.7) | 0.001 | 0.011 |
| Profit organization | 85,031 (66.8) | 16,145 (67.7) | 76,260 (66.8) | 0.018 | 0.006 |

*(Continued)*

**Table 1.** (Continued)

| Characteristic[a] | No. (%) of residents with concomitant antidepressant- opioid use | | | SDiff[b] | |
| | Overall sample (*n* = 127,200) | CYP2D6-inhibiting ADs with CYP2D6-metabolized opioids (*n* = 23,856)[c,d] | CYP2D6-neutral ADs with CYP2D6-metabolized opioids (*n* = 114,140)[c,d] | Before IPTW | After IPTW |
|---|---|---|---|---|---|
| Any special care unit | 30,163 (23.7) | 5,420 (22.7) | 27,158 (23.8) | 0.025 | 0.000 |
| Geography | | | | | |
| Metropolitan (RUCCs = 1–3) | 86,389 (67.9) | 15,769 (66.1) | 77,743 (68.1) | 0.043 | 0.025 |
| Micropolitan (RUCCs = 4–7) | 31,941 (25.1) | 6,361 (26.7) | 28,524 (25.0) | 0.038 | 0.022 |
| Rural (RUCCs = 8–9) | 8,870 (7.0) | 1,726 (7.2) | 7,873 (6.9) | 0.013 | 0.008 |
| **Year of cohort entry** | | | | | |
| 2011 | 16,660 (13.1) | 2,827 (11.9) | 14,606 (12.8) | 0.029 | 0.010 |
| 2012 | 14,966 (11.8) | 2,631 (11.0) | 13,240 (11.6) | 0.018 | 0.013 |
| 2013 | 14,965 (11.8) | 2,613 (11.0) | 13,318 (11.7) | 0.023 | 0.00 |
| 2014 | 14,458 (11.4) | 2,643 (11.1) | 12,914 (11.3) | 0.008 | 0.002 |
| 2015 | 13,451 (10.6) | 2,461 (10.3) | 12,147 (10.6) | 0.011 | 0.003 |
| 2016 | 13,630 (10.7) | 2,505 (10.5) | 12,379 (10.9) | 0.011 | 0.003 |
| 2017 | 12,380 (9.7) | 2,427 (10.2) | 11,207 (9.8) | 0.012 | 0.002 |
| 2018 | 11,053 (8.7) | 2,205 (9.2) | 10,097 (8.9) | 0.014 | 0.005 |
| 2019 | 7,239 (5.7) | 1,658 (7.0) | 16,575 (5.8) | 0.049 | 0.007 |
| 2020 | 8,398 (6.6) | 1,886 (7.9) | 7,657 (6.7) | 0.046 | 0.011 |

[a]Clinical characteristics were measured in the 6 months before cohort entry, that is, day 1 of concomitant use of antidepressants and CYP2D6-metabolized opioids.

[b]Covariates with SDiff higher than 0.100 represent meaningful differences between case and control groups.

[c]Values are percentages unless otherwise specified.

[d]A resident could contribute to more than 1 observation during the study period.

[e]Included Asian, Hispanic, Native American, and Pacific Islander.

Abbreviations: AD, antidepressants; ADL, activities of daily living; CNS, central nervous system; CYP, cytochrome P450; IPTW, inverse probability of treatment weighting; MME, morphine milligram equivalent; PHQ-9, Patient Health Questionnaire-9; PRN, as needed; RUCCs, rural-urban continuum codes; SDiff, standardized difference.

*P* = 0.53; RD, 0.2% [95% CI, −0.4%, 0.8%]). Interaction analyses of exposure and quarterly time showed that the associations for clinical outcomes did not statistically differ over time (S5 Table).

For ORAE outcomes, higher crude incidence rates (per 1,000 patient-years) for pain-related hospitalizations (15.37 versus 11.27), pain-related ED visits (8.32 versus 6.19), OUD (7.52 versus 5.61), and OD (5.84 versus 4.38) were observed in the study versus comparison group (Table 3). After covariate adjustment, the study versus comparison group had higher incidence rates of pain-related hospitalizations (aIRR, 1.13 [95% CI, 1.04, 1.22]; *P* = 0.003) and pain-related ED visits (aIRR, 1.17 [95% CI, 1.07, 1.29]; *P* = 0.003) during follow-up. The adjusted absolute increase in the incidence of pain-related hospitalizations and ED visits in the study versus comparison group were 1.12 cases per 1,000 patient-years (95% CI, 0.39, 1.89) and 0.85 cases per 1,000 patient-years (95% CI, 0.29, 1.41). Conversely, the study versus comparison group had no difference was observed for OUD (aIRR, 1.17 [95% CI, 0.84–1.63]; *P* = 0.34; RD, 0.83 [95% CI, −0.92, 2.58]) and OD between groups (aIRR, 1.21 [95% CI, 0.93–1.59]; *P* = 0.16; RD, 0.80 [95% CI, −0.36, 1.95]).

### Results of sensitivity and subgroup analyses

The stratification analysis by antidepressant classes with or without additional analgesic effects showed results consistent with the main analysis for all clinical outcomes, OUD, and OD adverse outcomes. (Table 4) However,

**Table 2. Associations of residents receiving concomitant use of CYP2D6-metabolized opioids with existing CYP2D6-inhibiting antidepressants (study group) vs. CYP2D6-neutral antidepressants (comparison group) with clinical worsening outcomes.**

| | Crude estimate | | Study group vs. comparison group | | | | Absolute standardized estimate | | |
|---|---|---|---|---|---|---|---|---|---|
| | Study group | Comparison group | | | | | Study group | Comparison group | Difference |
| Clinical outcomes[a] | No.[b] (%) | | Crude RR (95% CI) | P-value | Adjusted RR[c] (95% CI) | P-value | % (95% CI) | | |
| Worsening pain | 68,964 (25.5) | 313,587 (23.0) | 1.10 (1.08–1.12) | <0.001 | 1.04 (1.02–1.06) | <0.001 | 25.1 (24.6, 25.6) | 24.0 (23.8, 24.2) | 1.1 (0.6, 1.6) |
| Worsening physical function | 68,964 (63.8) | 313,587 (65.4) | 0.97 (0.96–0.98) | <0.001 | 1.00 (0.99–1.01) | 0.49 | 64.9 (64.3, 65.5) | 65.1 (64.8, 65.4) | −0.2 (−0.9, 0.5) |
| Worsening depression | 68,964 (29.6) | 313,587 (29.9) | 0.99 (0.97–1.00) | 0.14 | 1.01 (0.99–.1.03) | 0.53 | 31.1 (30.4, 31.6) | 30.8 (30.6, 31.0) | 0.2 (−0.4, 0.8) |

[a]Clinical outcomes were measured in a subset of the sample with at least one Minimum Data Set 3.0 in follow-up.

[b]Total number of resident quarters during the study period.

[c]Logistic regression model with a generalized estimating equation that adjusted for baseline covariates via the inverse probability of treatment weighting and quarter (time) as covariates for clinical outcomes.

Abbreviations: AD, antidepressants; CYP, cytochrome P450; CI, confidence interval; RR, rate ratio.

**Table 3. Associations of residents receiving concomitant use of CYP2D6-metabolized opioids with existing CYP2D6-inhibiting antidepressants (study group) vs. CYP2D6-neutral antidepressants (comparison group) with opioid-related adverse outcomes.**

| | Crude estimate | | Study group vs. comparison group | | | | Absolute standardized estimate | | |
|---|---|---|---|---|---|---|---|---|---|
| | Study group | Comparison group | | | | | Study group | Comparison group | Rate Difference |
| Adverse outcomes | Incidence rate per 1,000 PYs (event/PYs), No. | | Crude RR (95% CI) | P-value | Adjusted RR[a] (95% CI) | P-value | Incidence rate per 1,000 PYs (95% CI) | | |
| Pain-related hospitalization | 15.37 (303/19716), 23,856 | 11.27 (1020/90498), 114,140 | 1.36 (1.20, 1.55) | <0.001 | 1.13 (1.04, 1.22) | 0.003 | 10.24 (9.69, 10.78) | 9.10 (8.58, 9.62) | 1.12 (0.39, 1.89) |
| Pain-related ED visit | 8.32 (164/19716), 23,856 | 6.19 (560/90498), 114,140 | 1.34 (1.13, 1.60) | <0.001 | 1.17 (1.07, 1.29) | 0.003 | 5.81 (5.20, 6.22) | 4.96 (4.58, 5.34) | 0.85 (0.29, 1.41) |
| Opioid use disorder[b] | 7.52 (148/19684), 23,813 | 5.61 (507/90385), 113,993 | 1.35 (0.89, 2.03) | 0.16 | 1.17 (0.85, 1.63) | 0.34 | 5.65 (4.20, 7.10) | 4.82 (3.84, 5.80) | 0.83 (−0.92, 2.58) |
| Opioid overdose[b] | 5.84 (115/19684), 23,813 | 4.38 (157/90385), 113,993 | 1.37 (1.01,1.85) | 0.04 | 1.22 (0.93, 1.60) | 0.16 | 4.48 (3.46, 5.49) | 3.68 (3.13, 4.24) | 0.80 (−0.36, 1.95) |

[a]Poisson or negative binomial regression that adjusted for baseline covariates via the inverse probability of treatment weighting and total number of days in follow-up as an offset variable.

[b]Restricted to the sample with no diagnosis of opioid use disorder or overdose at baseline.

Abbreviations: AD, antidepressants; CYP, cytochrome P450; CI, confidence interval; ED, emergency department; IRR, incidence rate ratio; PY, patient-year; RR, rate ratio.

increased associations for pain-related hospitalizations (aIRR, 1.21 [95% CI, 1.11, 1.32]; P<0.001) and ED visits (aIRR, 1.26 [95% CI, 1.12, 1.41]; P<0.001) were observed only among CYP2D6-metabolized opioid users who concurrently used SSRIs or other antidepressants that had no additional analgesic effects. We also observed consistent results for clinical and ORAE outcomes among residents when stratified by specific CYP2D6-metabolized opioids (except for codeine likely due to a relatively small sample size) (S6 Table). The sensitivity analyses that accounted for censoring due to death (S7 Table), excluded use of other CYP2D6 drugs at baseline (S8 Table), and included only the first eligible resident observation (S9 Table) yielded results consistent with the main analysis for clinical and ORAE outcomes.

**Table 4. Unadjusted and adjusted associations of concomitant use of antidepressants and CYP2D6-metabolized opioids with clinical worsening and opioid-related adverse outcomes, stratified by antidepressant therapeutic classes with vs. without additional analgesic effects.**

| | Concomitant use of CYP2D6-inhibiting ADs (vs. CYP2D6-neutral ADs) with CYP2D6-metabolized opioids | | | | | | | |
|---|---|---|---|---|---|---|---|---|
| | Antidepressants with analgesic effect[b] (n = 15,585) | | | | Antidepressants without analgesic effec[b] (n = 99,079) | | | |
| Clinical Outcomes[a] | Crude RR (95% CI) | P-value | Adjusted RR[c] (95% CI) | P-value | Crude RR (95% CI) | P-value | Adjusted RR[c] (95% CI) | P-value |
| Worsening pain | 1.02 (0.98,1.06) | 0.30 | 1.03 (1.00,1.06) | 0.04 | 1.10 (1.08,1.13) | <0.001 | 1.05 (1.02, 1.07) | <0.001 |
| Worsening physical function | 1.00 (0.98,1.03) | 0.64 | 1.02 (0.99,1.04) | 0.14 | 0.98 (0.97, 0.99) | <0.001 | 1.00 (0.98, 1.01) | 0.45 |
| Worsening depression | 0.96 (0.92,0.99) | 0.03 | 0.99 (0.95,1.03) | 0.55 | 1.00 (0.98,1.03) | 0.75 | 1.01 (0.98, 1.03) | 0.53 |
| | Antidepressants with analgesic effect[b] (n = 18,558) | | | | Antidepressants without analgesic effect[b] (n = 119,438) | | | |
| Adverse outcomes | Crude IRR (95% CI) | P-value | Adjusted IRR[d] (95% CI) | P-value | Crude IRR (95% CI) | P-value | Adjusted IRR[d] (95% CI) | P-value |
| Pain-related hospitalization | 0.94 (0.72, 1.21) | 0.61 | 0.95 (0.79, 1.14) | 0.57 | 1.45 (1.24, 1.69) | <0.001 | 1.21 (1.11, 1.32) | <0.001 |
| Pain-related ED visit | 0.98 (0.69, 1.39) | 0.92 | 0.96 (0.74, 1.24) | 0.73 | 1.41 (1.14, 1.74) | 0.001 | 1.26 (1.12, 1.41) | <0.001 |
| Opioid use disorder[e] | 1.38 (0.65, 2.91) | 0.40 | 1.26 (0.60, 2.64) | 0.54 | 1.32 (0.79, 2.21) | 0.29 | 1.20 (0.84, 1.71) | 0.31 |
| Opioid overdose[e] | 1.56 (0.79, 3.09) | 0.20 | 1.16 (0.74, 1.81) | 0.51 | 1.27 (0.88, 1.82) | 0.20 | 1.25 (0.96, 1.63) | 0.10 |

[a]Clinical outcomes were measured in a subset of the sample with at least one MDS 3.0 in follow-up.

[b]Antidepressants with analgesic effect are norepinephrine reuptake inhibitors and tricyclic antidepressants; antidepressants without analgesic effect are selective serotonin reuptake inhibitors and other antidepressants.

[c]A robust Poisson regression model with a generalized estimating equation that adjusted for baseline covariates via the inverse probability of treatment weighting and quarter (time) as covariates for clinical outcomes.

[d]Poisson or negative binomial regression that adjusted for baseline covariates via the inverse probability of treatment weighting and total number of days in follow-up as an offset variable.

[e]Restricted to the sample with no diagnosis of opioid use disorder or opioid overdose at baseline.

Abbreviations: AD, antidepressants; CYP, cytochrome P450; IRR; incidence rate ratio; MDS, minimum data set; RR, rate ratio.

## Discussion

This cohort study using a 100% nursing home resident sample linked to their Medicare claims data is, to our knowledge, among the first to provide population-based data on the safety of initiating CYP2D6-metabolized opioids among older Medicare nursing home residents who already received antidepressants. We found that residents who initiated CYP2D6-metabolized opioids while receiving CYP2D6-inhibiting (versus residents receiving CYP2D6-neutral) antidepressants had a higher risk of worsening pain and higher incidence rates of pain-related hospitalizations and pain-related ED visits, with no difference in physical function, depression, OUD, and OD. Consistent findings were observed in sensitivity and multiple subgroup analyses. We observed small magnitude of RR (4%) and RD (1.1%) for worsening pain and moderate magnitude for pain-related hospitalizations (RR = 13%; RD = 1.21 per 1,000 patient-years) and ED visits (RR = 17%; RD = 0.85 per 1,000 patient-years). These findings suggest that initiating CYP2D6-metabolized opioids on existing antidepressants that strongly or moderately inhibit the CYP2D6 enzyme was associated with worsening pain control and increased risk for pain-related medical encounters, although their RRs and RDs appear to be small to moderate during the follow-up period (up to 1 year) among older nursing home residents.

To our best knowledge, only one prior population-based study focused on older nursing home residents and examined the safety of opioid-antidepressant interaction triggered by initiation of antidepressants (i.e., antidepressant-triggered interaction) [8]. Consistent with the prior study, the present study that examined the interaction triggered by initiation of CYP2D6-metabolized opioids (i.e., opioid-triggered interaction) observed increased risk of worsening pain, pain-related hospitalizations, and pain-related ED visits for residents with CYP2D6-inhibiting (versus CYP2D6-neutral) antidepressants

concomitantly used with CYP2D6-metabolized opioids. The finding for worsening pain echoes the results of other prior studies conducted in patients with depression or surgery in a single clinical setting [4,5], and results of clinical trials [6,7]. While the direction of associations for these adverse outcomes is similar, the magnitude of these associations appears to differ, with smaller relative and absolute effects on worsening pain (RR = 1.04 versus 1.13; RD = 1.1% versus 3.9%), pain-related hospitalization (IRR = 1.13 versus 1.37; RD = 1.12 versus 3.96 per 1,000 patient-years), and pain-related ED visit (IRR = 1.17 versus 1.49; RD = 0.85 versus 3.02 per 1,000 patient-years) associated with the opioid-triggered interactions versus antidepressant-triggered interactions [8]. The difference in association magnitude supports our hypothesis that the degree of impact on adverse outcomes associated with opioid-antidepressant interactions varies by the sequence of drug initiation.

The present study, however, did not observe significant associations for OUD, in contrast to the prior study that investigated opioid-antidepressant interaction triggered by initiation of antidepressants among older nursing home residents [8]. The disparate finding between the prior and present study is likely explained by the difference in the sequence of opioid or antidepressant use. The prior study focused on existing users of CYP2D6-metabolized opioids who likely developed opioid dependence at baseline. Initiating CYP2D6-inhibiting antidepressants on existing CYP2D6-metabolized opioids decreased opioid analgesic effect, leading to increasing use of opioids to relieve pain or inability to reduce opioid use, symptoms considered for the diagnosis of OUD [36]. Conversely, the present study focused on existing users of antidepressants who had not yet received opioids at baseline. Given that this group of patients was at no risk for opioid dependence, initiating CYP2D6-metabolized opioids on existing antidepressants, although decreasing opioid analgesic effect, had no association with OUD. Consistent with the prior study [8], the present study found no association with physical function, depressive symptoms, and OD among older nursing home adults, suggesting that these patient outcomes may not be immediately affected by concomitant antidepressant-opioid use regardless of the sequence of drug initiation. No difference in OD was observed between the study and comparison group, which is consistent with preclinical [37,38] and pharmacokinetic modeling studies [39,40], suggesting that use of CYP2D6 inhibitors alone with oxycodone is not associated with increased oxycodone and oxymorphone concentration, a major underlying cause of OD.

The present study provides referential data for clinicians regarding the safety of initiation of CYP2D6-metabolized opioids on existing antidepressants (i.e., opioid-triggered interaction), which was predominantly used by older nursing home residents for treating their comorbid depression and chronic pain. Among patients whose opioid-antidepressant interaction was triggered by the opioids, use of CYP2D6-metabolized opioids concomitantly with CYP2D6-inhibiting (versus CYPD2D6-neutral) antidepressants was associated with an increased risk of worsening pain and risk of pain-related hospitalizations and ED visits. However, the relative and absolute risks of these outcomes associated with opioid-triggered interactions appear to be low to moderate, compared to the estimates associated with antidepressant-triggered interactions [8]. Overall, our study along with the prior study [8] suggests that when co-use of CYP2D6-metabolized opioids and antidepressants is clinically needed, selection of CYP2D6-neutral antidepressants, rather than CYP2D6-inhibiting antidepressants, may reduce the risk of worsening pain and pain-related hospital and ED visits.

## Limitations

This study has limitations. First, Medicare prescription drug event data provide information on prescription drugs dispensed but not consumed and do not capture self-paid prescriptions or medications covered by non-Medicare programs. For example, 6% of Medicare beneficiaries were veterans in 2022 [41] and may obtain their prescriptions through veterans' insurance program [42]. Second, Medicare data do not provide genetic information about individuals' capacity for metabolizing opioids and antidepressants, which may affect adverse outcomes. Lack of individual genetic information also prevents us from examining the heterogeneity of effects on adverse outcomes associated with opioid-antidepressant interactions across subgroups by CYP2D6 genotypes. Nevertheless, the present study provides average effects in the older nursing home population who initiated CYP2D6-metabolized opioids while receiving antidepressants. Third,

Medicare Part D data provided no information on disease indications for which a drug is prescribed. To address this issue, we required eligible residents to have a diagnosis of chronic pain (for opioid initiation) and a diagnosis of indications FDA-approved or off-label use (for antidepressant treatment) during the baseline period to minimize potential confounding by indication. Fourth, because we relied on resident reports for clinical outcomes and administrative claims data for pain-related hospitalizations and ED visits, measurement bias was possible, although such bias is likely nondifferential between the study and comparison groups. Fifth, the underdiagnosis of OUD was possible, but this too is likely nondifferential between groups. Sixth, while we accounted for many potential confounders, our results remain subject to unknown or unmeasured confounding. Seventh, we used intention-to-treat analysis without censoring patients when they stopped CYP2D6-metabolized opioids or antidepressants or when they switched from CYP2D6-inhibiting to CYP2D6-neutral antidepressants (or vice versa) during the follow-up, which may yield conservative effect estimates, compared to per protocol analysis. Finally, our results can be generalized only to older nursing home long-term residents.

## Conclusions

In this cohort study of older nursing home residents who received antidepressants, concomitant use of CYP2D6-inhibiting (versus CYP-neutral) antidepressants with newly-prescribed CYP2D6-metabolized opioids was associated with increased risk of worsening pain and hospital and ED visits due to pain, although the relative and absolute risks of these outcomes are small to moderate. No association was found for physical function, depressive symptoms, OUD, and OD. The findings, consistent with a prior study [8], suggested that clinicians should be aware of potential worsening pain and hospital and ED visits due to pain among patients who concomitantly used CYP2D6-metabolizing opioids and antidepressants, particularly those with CYP2D6-inhibiting antidepressants. Caution is needed in interpreting our finding given the limitations of the data and nature of the observational study, which is subject to unmeasured confounding.

## Supporting information

**S1 STROBE Checklist. Strengthening the reporting of observational studies in epidemiology (STROBE) checklist.**
(DOC)

**S1 Text. Pre-specified analytical protocol.**
(DOCX)

**S1 Table. Medications of interest considered in the study.**
(DOCX)

**S2 Table. *ICD-9-CM, ICD-10-CM,* or procedure codes for disease conditions and service care considered in the study.**
(DOCX)

**S3 Table. Study covariates, definitions, and measurement sources and windows.**
(DOCX)

**S4 Table. Clinical and demographic characteristics of eligible patients with at least 1 MDS 3.0 in follow-up and received CYP2D6-opioids concomitantly with CYP2D6-inhibiting versus CYP2D6-neutral antidepressants.**
(DOCX)

**S5 Table** Quarterly associations of concomitant use of CYP2D6-metabolized opioids and antidepressants with clinical worsening outcomes from baseline to follow-up.
(DOCX)

**S6 Table. Unadjusted and adjusted associations of concomitant use of CYP2D6-metabolized opioids and antidepressants with clinical worsening and opioid-related adverse outcomes, stratified by specific CYP2D6-metabolized opioids.**
(DOCX)

**S7 Table. Associations of concomitant use of CYP2D6-metabolized opioids and antidepressants with clinical worsening and opioid-related adverse outcomes, adjusting for censoring due to death via inverse probability of censoring weighting.**
(DOCX)

**S8 Table. Sensitivity analysis of including eligible residents without use of other CYP2D6 medications at baseline.**
(DOCX)

**S9 Table** Sensitivity analysis of including eligible residents' first observation only.
(DOCX)

## Author contributions

**Conceptualization:** Yu-Jung Jenny Wei, Almut G. Winterstein, Roger B. Fillingim, Steven T. DeKosky, Stephan Schmidt.

**Data curation:** Yu-Jung Jenny Wei.

**Formal analysis:** Yu-Jung Jenny Wei, Michael J. Daniels.

**Funding acquisition:** Yu-Jung Jenny Wei, Almut G. Winterstein, Siegfried Schmidt, Roger B. Fillingim, Michael J. Daniels, Steven T. DeKosky, Stephan Schmidt.

**Investigation:** Yu-Jung Jenny Wei, Siegfried Schmidt, Roger B. Fillingim, Michael J. Daniels, Steven T. DeKosky, Stephan Schmidt.

**Methodology:** Yu-Jung Jenny Wei, Almut G. Winterstein.

**Project administration:** Yu-Jung Jenny Wei.

**Resources:** Yu-Jung Jenny Wei.

**Software:** Yu-Jung Jenny Wei.

**Supervision:** Yu-Jung Jenny Wei.

**Validation:** Yu-Jung Jenny Wei, Almut G. Winterstein, Siegfried Schmidt, Roger B. Fillingim, Michael J. Daniels, Steven T. DeKosky, Stephan Schmidt.

**Visualization:** Yu-Jung Jenny Wei.

**Writing – original draft:** Yu-Jung Jenny Wei.

**Writing – review & editing:** Yu-Jung Jenny Wei, Almut G. Winterstein, Siegfried Schmidt, Roger B. Fillingim, Michael J. Daniels, Steven T. DeKosky, Stephan Schmidt.

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
