## [Editor Report · Decision Letter 0]

12 Nov 2024

Dear Dr Wei, 

Thank you for submitting your manuscript entitled "Association of Initiating CYP2D6-metabolized Opioids with Risks of Adverse Outcomes in Older Adults Receiving Antidepressants—A Retrospective Cohort Study" for consideration by PLOS Medicine.

Your manuscript has now been evaluated by the PLOS Medicine editorial staff and I am writing to let you know that we would like to send your submission out for external peer review.

Please re-submit your manuscript within two working days, i.e. by Nov 14 2024 11:59PM. However, please do not hesitate to let us know if you need more time.

Kind regards,

Syba

Syba Sunny, MBBS, MRes, FRCPath

Associate Editor

PLOS Medicine

ssunny@plos.org

---

## [Decision Letter · Decision Letter 1]

27 Feb 2025

Dear Dr Wei,

Many thanks for submitting your manuscript "Association of Initiating CYP2D6-metabolized Opioids with Risks of Adverse Outcomes in Older Adults Receiving Antidepressants—A Retrospective Cohort Study" (PMEDICINE-D-24-03810R1) to PLOS Medicine. The paper has been reviewed by subject experts and a statistician; their comments are included below and can also be accessed here: [LINK]

As you will see, the reviewers raised some questions including around the inclusion criteria and the clinical relevance of the results. After discussing the paper with the editorial team, I'm pleased to invite you to revise the paper in response to the reviewers' comments. We plan to assess the revision, and will then determine if we can send the revised paper to all of the original reviewers. We cannot provide any guarantees at this stage regarding publication.

We ask that you submit your revision by Mar 20 2025 11:59PM. However, if this deadline is not feasible, please contact me by email, and we can discuss a suitable alternative.

Don't hesitate to contact me directly with any questions (rkirk@plos.org). 

Best regards, 

Rebecca 

Rebecca Kirk, 

PLOS Medicine

rkirk@plos.org

Comments from the reviewers: 

Reviewer #1: This study aimed to identify whether initiating CYP2D6-metabolised opioids in older adults who were already taking CYP2D6-inhibiting antidepressants was associated with adverse outcomes. This extended earlier research that found in a similar patient population who were already taking CYP2D6-metabolised opioids, initiating CYP2D6-inhibiting antidepressants was associated with adverse outcomes including increased pain. The study design was a retrospective cohort study utilising USA Medicare nursing home data and controlled for potential baseline imbalance with inverse probability of treatment weighting.

The manuscript is clear and well written. The authors effectively establish the importance of this research, situate it with respect to the existing literature, and define the characteristics of the cohort and data sources well. I also commend their proactive use of the STROBE checklist. While I suspect their analytical approach is broadly sound, I have some concerns and further detail is needed, so this is the focus of my feedback. Also, while the Discussion is very well constructed and argued, the authors don't appear to consider clinical significance (in addition to statistical significance), which is especially important when analysing large volumes of data. 

Detailed feedback on the manuscript is provided below. My primary focus has been methodological, but I have made a few broader points as well. All items are major unless indicated otherwise.

1. [MINOR] I understand there is a lot to squeeze into the abstract but the fact all analyses are based on IPTW is important to know. I'd encourage the authors to find some way to identify this in either the abstract or, alternatively, the title. 

2. What is the difference between a "nursing home" and a "skilled nursing facility" and what was the reasoning behind omitting patients with a stay at the latter? This excluded a substantial number of patients, and the reasoning isn't immediately clear to non-US readers. Did you consider sensitivity analyses including these patient (or separate analyses exclusively on these patients)?

3. Also with respect to inclusion/exclusion, Fig 1 says ~70k patients were excluded because they did not have a "… diagnosis of chronic pain for CYP2D6-metabolised opioids during 1 month before cohort entry", but I did not see this in the methods. Suggest making this clearer in the body of the manuscript given the large number of excluded patients. 

4. "Residents were followed up from cohort entry until the end of 1-year follow-up, Medicare disenrollment, death, NH discharge, or study end (December 31, 2021)." Two things. Firstly, suggest the distribution of these outcomes is reported. Secondly, why did the authors enrol patients in 2021 with the knowledge they would not have a full year of follow-up? That is, why not cease enrolling new participants at the end of 2020 to ensure all enrolled patients could have had a full year of follow-up? One risk here is the influence of unobserved confounders that vary over time. 

5. "Residents were allowed to re-enter the cohort after the end of follow-up if they met the eligibility criteria." How did the authors account for this non-independence in their statistical analyses? 

6. Relatedly, I note that the authors incorporated some site-level covariates, but I don't think the site/patient/observation hierarchy was incorporated into the statistical modelling. Please clarify and consider the risk of site-level variation not adequately captured in the included site-level covariates. 

7. Please describe how the propensity score was calculated and include an appropriate reference. 

8. Please justify your choice of IPTW over other propensity score-based methods (such as propensity score matching) and include an appropriate reference. Note I am not suggesting the authors shouldn't have used IPTW, rather it is one option of several with different pros and cons so understanding the reasoning is important. 

9. "For each clinical outcome, we used a modified Poisson regression…" How was it modified and why? I suspect you mean robust Poisson regression was used but this needs clarification and an appropriate reference. 

10. "For ORAE count outcomes, we used a Poisson or negative binomial model…" How was this decision made? I suspect based on the presence of overdispersion but important to make clear. 

11. I don't understand why reference [20] was included. The cited literature examined zero-inflated mixture models which are not employed here. 

12. Similarly, I don't understand why reference [21] was included. The cited literature examined IPCW but as a win ratio statistic which is not used here. Further clarity here is important - what problem are you trying to solve and why is your choice of approach the optimal solution? 

13. I am concerned the clinical significance of the outcomes is not sufficiently considered (separately to the statistical significance), which is a risk with large sample sizes like in this research. For example, the primary result "the study vs. comparison group had higher risk of worsening pain (aRR, 1.04 [95% CI, 1.02-1.06]; P < .001)…" is undoubtedly statistically significant but is a 4% difference between groups of clinical importance? Also, when comparing this study and the authors' earlier study it is suggested "…the magnitude of these associations differs, with a smaller effect on worsening pain (RR=1.04 vs 1.03)…"; here I would question a 4% worsening of pain is "different in magnitude" to a 3% worsening of pain. Suggest discussing clinical significance ideally referencing reasonable minimum differences established clinically, and perhaps moderating language in some cases. 

14. [MINOR] "Our observed null association with OD is also supported by pharmacokinetic…" Be careful with wording here - absence of evidence is not evidence of absence. 

15. [MINOR] Considered together, the main publication and supplementary materials report a very large number of p-values with the attendant risk of Type I error(s). I appreciate they are included for completeness however I would encourage consideration of omitting some - particularly in the supplementary materials, noting 95% CIs suffice in many cases. 

16. Given the complexity of the data management and statistical analyses, I would strongly encourage the authors to include their code to facilitate understanding and replicability. At a minimum this would include their statistical modelling code and, ideally, code developed to produce variables of interest including outcomes, exposures, and other covariates. 

Reviewer #2: This large cohort study utilized nationwide Medicare data from the USA to compare clinical outcomes among older nursing home residents who initiated CYP2D6-metabolized opioids while already using CYP2D6-inhibiting versus CYP2D6-neutral antidepressants between 2010 and 2021. The authors followed individuals for one year and analyzed three distinct outcome groups: pain intensity, physical functioning, and depression. After adjusting for a wide range of potential confounders and covariates using inverse probability of treatment weighting, the study found that users of CYP2D6-inhibiting antidepressants were associated with worsened pain-related outcomes, including increased rates of pain-related hospitalizations and emergency department (ED) visits. However, no significant differences were observed between the groups in physical functioning or depressive symptoms.

This study is well-designed and structured, leveraging exceptionally large-scale data to address an important research question about the vulnerable older population. The authors effectively controlled for confounding variables using robust statistical methods and conducted a wide range of sensitivity analyses. The manuscript is well-written, with a balanced presentation of the relevant literature. Moreover, the conclusions are well-supported by the data and results. However, the current reviewer has outlined the following queries, requests for clarification, and suggested revisions to further enhance the manuscript's quality:

1. Introduction: While the authors discuss the potential mechanisms underlying interactions between these drug groups and their impact on outcomes to some extent, further elaboration on these mechanisms would strengthen the study's background and justification. This would help establish clearer hypotheses for the three outcomes: worsening depressive symptoms, impaired physical functioning, and increased pain intensity.

2. Methods - Covariates: Why was the calendar year not included as a covariate? Prescribing patterns for opioids and antidepressants, as well as clinical guidelines and care facility utilization, likely changed over time and could influence the outcomes.

3. Sensitivity Analyses: It would be valuable to perform an additional analysis restricting antidepressant users to those with a treatment indication of depression. Since antidepressants in older adults may also be prescribed for non-depressive indications (e.g., pain), excluding individuals with such indications could help clarify and rule out the possibility that worsening pain intensity was due to selection bias—specifically, the inclusion of individuals already using antidepressants for pain who may have experienced greater worsening in pain intensity.

4. If data on treatment indications for antidepressants and opioids are unavailable, this limitation should be acknowledged, as confounding by indication is a significant concern in observational research.

5. Sensitivity Analyses - Clarification: The sensitivity analyses section would benefit from further explanation of the purpose of each sensitivity analysis and what potential biases or issues they aim to address. This may not be immediately clear to all readers.

6. Discussion: In the first paragraph of the discussion section, the authors state that "CYP2D6 enzyme led to worsening pain control." The phrase "led to" implies causality, which is inappropriate in this context. The statement should be revised to avoid suggesting causation.

7. Public Health Impact: To enhance the discussion of public health implications, the authors could calculate an attributable fraction for the population. This would estimate how many excess cases of hospitalizations or ED visits could be prevented by prescribing CYP2D6-neutral antidepressants alongside CYP2D6-metabolized opioids.

8. Limitations: Regarding the first limitation, how large is the group of individuals whose prescriptions were self-paid or covered by non-Medicare programs? Including this information would provide additional context.

9. Prescriber Information: Is there any data on prescriber qualifications (e.g., general practitioners, psychiatrists, or other specialists)? If prescriber characteristics might influence drug selection and treatment outcomes, this limitation should be discussed.

10. Tables: The presentation of findings in Tables 2 and 3 could be improved by using forest plots, which would make the results more visually engaging and easier to interpret compared to the current monotone tabular format.

Reviewer #3: Thank you so much for the opportunity to review this interesting study. 

Overall, I think the research question is articulated clearly. The prose is concise and well written. However I don't believe the results are robust or are meaningful enough to have clinical relevance. I have a number of serious concerns surrounding the study. 

1) First, I am concerned that the CYP2D6 inhibiting antidepressants were not classified appropriately. For example, some of the medications listed in the CYP2D6 neutral group actually have CYP2D6 inhibiting properties, albeit weaker than paxil and prozac (i.e., sertraline, desvenlafaxine, citalopram). If duloxetine (a mild to moderate 2D6 inhibitor) was grouped with paxil and prozac, shouldn't you group other mild or moderate 2D6 inhibitors?

2) There is no doubt that 2D6 metabolism is impacted by various antidepressants. Whether this is a clinically meaningful impact is a whole different story. In short, I am not convinced the authors established the theoretical foundation for focusing on 2D6

Some recent studies have suggested that 2D6 inhibition itself is inadequate to meaningfully interfere with oxycodone metabolism. See PMID 20857093 for more details: "paroxetine pretreatment inhibits cyp2d6 without inducing relevant changes in oxycodone exposure and partially blunts the pharmacodynamic effects of oxycodone due to intrinsic pharmacological activities." Also see PMID 21142269 "the effect of paroxetine on plasma concentrations of oxycodone was negligible." 

Overall, I am not convinced that the association observed here is even driven by CYP2D6 effects. Namely, several of the antidepressants in the 2D6 group are well known to be anticholinergic and even land on the Beers list of geriatric medications. How do we know it's the 2D6 that's the underlying cause? 

3) In clinical practice, cymbalta and paxil also tend not to be 1st line meds for bread and butter depression. This raises questions about unmeasured confounding. Could it just be that the people receiving paxil and cymbalta are sicker at baseline -- and have more comorbidities -- than the other group, contributing to the observed association? 

I appreciate the use of IPTWs w/ Standard diffs showing good balance between the groups. However, I am still concerned about residual confounding (i.e., there was no adjustment for the amount of time people were on opioids, the dosage of opioids, the type of opioids, etc.) This could very well confound the observed association. The fact that the study group had 61.6% using an adjuvant analgesic vs 37% in the control group also makes me wonder if the study group just had more baseline pain levels and would have had higher pain related healthcare encounters no matter what antidepressant. 

4) The results don't appear very robust or clinically meaningful enough to justify the authors' conclusions: For example, an 15% vs 11% difference in pain related hospitalization, with a confidence interval of 1.04-1.21, is a very small difference. Should a clinician switch someone who has been stable on prozac to something else when the difference is merely 15% vs 11% (w/ a 95%CI abutting 1.00)? Several of the secondary analyses seemed to have no association between 2D6 inhibitors vs 2D6 neutral meds and the outcome variables of interest. 

Minor

5) I'm a little confused why you required residents to have coma or severe cognitive impairment - this could confound the pain assessment and the reasons for hospitalizations. Pain assessment would have much higher variability for a person with cognitive impairment. If it's being assessed by different providers, there would also be more operator variability. This can greatly skew your results. 

6) Pain related hospitalization and ED visits were defined as encounters with a primary or secondary diagnosis of a chronic pain condition: there is a lot of room for error here. Lots of chronic pain is coded as a hospital problem, even if the plan is to just continue home medications. It does not necessarily mean that the healthcare encounter was at all a result of the pain. There's no way to know that the pain was current and pertinent to the admission or ED visit. 

7) On page 9, you say we classified residents with score changes equal to 0 that remained moderate to severe as having worsening outcomes. Similarly, you classified residents with score change of 0 that had no-to-mild pain as improving outcome. Those two scenarios are by definition the same outcome, not worsening or improving outcomes? 

8) On page 6, you say among older NH patients who concomitantly use both CYP2D6-metabolized opioids and antidepressants, 80% are those who received antidepressants first followed by opioids. What is the source of this statistic? 

Reviewer #4: The authors provide a very well written manuscript investigating the addition of opioids to a treatment regimen of antidepressants and its consequences. The outcomes are informative and appropriately chosen for purpose in clinical practice, which is a particular strength of the research. 

I have only a few questions or suggestions:

1. The methods are generally well chosen considering interdependency of individuals included several times and repeated outcome measurements and covariate balance and adjustment. However, re-entering of previous users may dilute the effect due to depletion of susceptibles, even though the influence of drug-drug interactions may be virgin. However due to the complexity of IA depletion of susceptibles may still be relevant? Please comment on this potential bias.

2. According to the authors one day of overlap was sufficient to qualify for concurrent use. The authors allowed gaps of less than 15 days between precriptions. Please detail how episodes of concurrent exposure were constructed and how this may effect the risk estimates during the one year of follow-up. This is imprtnat to be reported to enable reproduceability of results. 

3. Would any information of dosage of drugs be availbale for the analysis and how could this consideration have affected the findings?

4. The authors consider concomitant use of other CYP inhibitors and inducers at baseline and by exclusion in sensitivity analyses. a) For reproducabiliyty reasons, I could not find any information of how these were identified and which drugs fell into which category. b) Moreover, it would have been interesting to explore the association between the burden of inhibitors and inducers on the outcomes during follow-up in a time varying manner, but also depending on how concurrent use episodes of the opioid drugs of interest were constructed. It is not clear why the other drugs metabolized by CYP would have more or less impact than the antidepressants under investigation, in particular if used in a regular manner. Please comment on that. 

5. Communication of absolute risk differences and number needed to harm could be clincally relevant putting the reported 4% higher risk of 

worsening pain and 13% and 17% higher incidence rates of pain-related hospitalizations and pain-related ED visits into perspective.

---

* Please upload any figures associated with your paper as individual TIF or EPS files with 300dpi resolution at resubmission; please read our figure guidelines for more information on our requirements: http://journals.plos.org/plosmedicine/s/figures. While revising your submission, please upload your figure files to the PACE digital diagnostic tool, https://pacev2.apexcovantage.com/. PACE helps ensure that figures meet PLOS requirements. To use PACE, you must first register as a user. Then, login and navigate to the UPLOAD tab, where you will find detailed instructions on how to use the tool. If you encounter any issues or have any questions when using PACE, please email us at PLOSMedicine@plos.org.

FIGURES AND TABLES

SUPPLEMENTARY MATERIAL

REFERENCES

OBSERVATIONAL STUDIES

* Abstract: Please include the study design, population and setting, number of participants, years during which the study took place (enrollment and follow up), length of follow up, and main outcome measures.

* Please ensure that the study is reported according to the STROBE (or appropriate STOBE extension) guideline (available from: https://www.equator-network.org/reporting-guidelines/strobe) and include the completed STROBE (or STROBE extension) checklist as Supporting Information. Please add the following statement, or similar, to the Methods: "This study is reported as per the Strengthening the Reporting of Observational Studies in Epidemiology (STROBE) guideline (S1 Checklist)." When completing the checklist, please use section and paragraph numbers, rather than page numbers. 

* For all observational studies, in the manuscript text, please indicate: (1) the specific hypotheses you intended to test, (2) the analytical methods by which you planned to test them, (3) the analyses you actually performed, and (4) when reported analyses differ from those that were planned, transparent explanations for differences that affect the reliability of the study's results. If a reported analysis was performed based on an interesting but unanticipated pattern in the data, please be clear that the analysis was data driven. 

* Please state in the Methods section whether the study had a prospective protocol or analysis plan. If a prospective analysis plan (from your funding proposal, IRB or other ethics committee submission, study protocol, or other planning document written before analyzing the data) was used in designing the study, please include the relevant document(s) with your revised manuscript as a Supporting Information file to be published alongside your study and cite it in the Methods section. A legend for this file should be included at the end of your manuscript. If no such document exists, please make sure that the Methods section transparently describes when analyses were planned, and when/why any data-driven changes to analyses took place. Changes in the analysis, including those made in response to peer review comments, should be identified as such in the Methods section of the paper, with rationale.

---

## [Decision Letter · Decision Letter 2]

25 Apr 2025

Dear Dr. Wei,

Thank you very much for re-submitting your revised manuscript "Association of Initiating CYP2D6-metabolized Opioids with Risks of Adverse Outcomes in Older Adults Receiving Antidepressants—A Retrospective Cohort Study" (PMEDICINE-D-24-03810R2) for review by PLOS Medicine.

I have discussed the paper with my colleagues and the academic editor, and it was also seen again by four reviewers. As you will see, the reviewers were happy with your responses to their initial comments and the changes made to the manuscript. They raised a few additional questions, for example around the clinical implications of the findings, which can be addressed by including some additional discussion points and expanding on the limitations of the data. I am pleased to say that provided the manuscript is suitably revised in response to the reviewer comments and the remaining editorial and production issues are dealt with, we are planning to accept the paper for publication in the journal.

[LINK]

When preparing your revision, please once again ensure you address the specific points made by each reviewer and the editors. In your rebuttal letter you should indicate your response to the reviewers' and editors' comments and the changes you have made in the manuscript. Please submit a clean version of the paper as the main article file. A version with changes marked must also be uploaded as a marked up manuscript file.

We look forward to receiving the revised manuscript by May 02 2025 11:59PM. 

Sincerely,

Rebecca Kirk, 

Senior Editor 

PLOS Medicine

plosmedicine.org

Requests from Editors:

1. Please confirm that your abstract complies with our requirements, including providing all the information relevant to this study type https://journals.plos.org/plosmedicine/s/submission-guidelines#loc-abstract

2. Abstract, line 6-7 (minor): add “about”; eg “However, little is known *about* whether and to what extent…”

1. Please ensure that all abbreviations are defined at first use throughout the text. We prefer that you do not abbreviate “nursing home” throughout the text. 

2. Please confirm that all numbers presented in the abstract are present and identical to numbers presented in the main manuscript text.

3. Please review your text for claims of novelty or primacy (e.g. 'for the first time') and remove or qualify this language. For example, at line 348, you should qualify the statement by inserting the phrase “to our knowledge”. E.g., “This cohort study using a 100% nursing home resident sample linked to their Medicare claims data is, to our knowledge, among the first to provide population-based data…”

4. In the last sentence of the Abstract Methods and Findings section, please describe the main limitation(s) of the study's methodology.

5. In the author summary, in the final bullet point of 'What Do These Findings Mean?', please include the main limitations of the study in non-technical language.

6. Please state that your study had a prospective statistical analysis plan early in the Methods section, with a call-out to the location of this information in the supplemental information. 

7. In your response to the reviewers, you note that you provide examples of SAS codes for the statistical modelling in your paper in the SAP. Please include some text in the Methods/Statistical analysis section that points readers to the supplement for these examples. 

8. A reminder that all authors must declare their relevant competing interests per the PLOS policy, which can be seen here: https://journals.plos.org/plosmedicine/s/competing-interests For authors with ties to industry, please indicate whether any of the interests has a financial stake in the results of the current study.

Comments from Reviewers:

Reviewer #1: 

Thanks to the authors for their thoughtful and comprehensive responses to my review comments. I am satisfied the authors have addressed all my concerns and I am happy to recommend acceptance. 

I also wanted to include a few additional thoughts for the authors' future consideration. I don't think these are necessary for this particular paper because, fortuitously, there were small numbers of patients re-entering the cohort and limited clustering effects. That said, it would be remiss of me to not to be clear on these points for future cases when there is a pronounced lack of independence within the data. 

For items 5 and 6 the authors made use of PROC GLIMMIX to fit models accounting for non-independence within the data. The authors then experienced issues with convergence - a problem I have regularly encountered when fitting GLMMs to large datasets. I draw the authors' attention to a SAS paper I have found useful ( https://support.sas.com/resources/papers/proceedings18/2179-2018.pdf ) and a section of the SAS documentation that is important and not at all easy to find ( https://documentation.sas.com/doc/en/statug/15.2/statug_introcom_sect054.htm ). I hope the authors find these resources helpful when experiencing convergence issues in the future. The authors could also have considered using a GEE-type model ( https://documentation.sas.com/doc/en/pgmsascdc/9.4_3.3/statug/statug_genmod_examples05.htm ) as the estimates of interest are population-level, and GEEs tend to fit much easier than mixed models. As I said above, in this particular paper I am satisfied that approaches taken are sufficient and appropriate. However, this would not always be the case, so I wanted to ensure I provided some useful resources for future work. 

I also wanted to note that the authors were quite right in their response to item 15, and I thank them for drawing this to my attention. I agree the authors are compliant with the current editorial guidelines, and I have raised my thoughts on this with the editorial team. 

Thanks for the opportunity to review and congratulations on an excellent paper. 

Reviewer #2: 

Thank you for your thorough responses and the revisions made to the manuscript. I appreciate your efforts to address my concerns, particularly regarding the inclusion of adjusted absolute risks and risk differences, as well as the clarification on the limitations of the data.

While I still have some residual concerns regarding potential confounding by indication and the lack of study-specific data for certain limitations, I recognize that you have made a genuine effort to address these points. I acknowledge the inherent limitations of observational studies and believe that the remaining aspects are unlikely to significantly alter the overall findings.

Therefore, I strongly recommend that you:

Emphasize the need for caution when interpreting the results in light of the potential confounding by indication, particularly by adding a statement to the conclusion section that acknowledges this potential bias.

I believe that a thorough and transparent discussion of these limitations is crucial for the accurate interpretation of your findings. I recommend acceptance of the manuscript, with these minor revisions.

Reviewer #3: 

The authors have appropriately acknowledged the limitations of their work and were responsive to my comments. The remaining thing I struggle with is how physicians who prescribe opioids and antidepressants should interpret this data. At present, discussion of clinical implications is largely absent. 

My main question is: Do these results suggest that the opioid + CYP2D6 inhibitor combination is potentially harmful in a clinically meaningful way in older adults in nursing homes? Should clinicians reading this paper consider avoiding such co-prescriptions? 

The risks of any medication (i.e., opioids and antidepressants in this case) must be weighed against their potential benefits and alternatives. The effect sizes for many of the risks of opioid and CYP2D6 inhibitor co-prescribing were very small (i.e., RR of 1.04 for pain-related outcomes), and data on potential benefits of co-prescribing was outside the scope of the paper (and thus absent). 

If an older adult in a nursing home presents with opioids co-prescribed with CYP2D6 inhibiting antidepressants, I think it would be a stretch to suggest that the data here lend strong support for trying to get patients off the co-prescription. 

In conclusion, the data may be statistically significant, but not convincingly clinically significant. 

Reviewer #4: 

Thank you for answering my comments and providing additional detailed information on the methods, adjusted risk differences and attenuating the interpretation of the findings increasing the credibility of these kind of studies linking clinical pharmacology with population-based perspectives.

[LINK]

---

## [Editor Report · Decision Letter 3]

28 Apr 2025

Dear Dr Wei, 

On behalf of my colleagues and the Academic Editor, Louisa Degenhardt, I am pleased to inform you that we have agreed to publish your manuscript "Association of Initiating CYP2D6-metabolized Opioids with Risks of Adverse Outcomes in Older Adults Receiving Antidepressants—A Retrospective Cohort Study" (PMEDICINE-D-24-03810R3) in PLOS Medicine.

PRESS

Sincerely, 

Rebecca Kirk 

Senior Editor 

PLOS Medicine